# Using of n-grams from morphological tags for fake news classification



Jozef Kapusta[1], Martin Drlik[1] and Michal Munk[1,2]

[1] Department of Informatics, Constantine the Philosopher University in Nitra, Nitra, Slovakia
[2] Science and Research Centre, University of Pardubice, Pardubice, Czech Republic

## ABSTRACT

Research of the techniques for effective fake news detection has become very needed and attractive. These techniques have a background in many research disciplines, including morphological analysis. Several researchers stated that simple content-related n-grams and POS tagging had been proven insufficient for fake news classification. However, they did not realise any empirical research results, which could confirm these statements experimentally in the last decade. Considering this contradiction, the main aim of the paper is to experimentally evaluate the potential of the common use of n-grams and POS tags for the correct classification of fake and true news. The dataset of published fake or real news about the current Covid-19 pandemic was pre-processed using morphological analysis. As a result, n-grams of POS tags were prepared and further analysed. Three techniques based on POS tags were proposed and applied to different groups of n-grams in the pre-processing phase of fake news detection. The n-gram size was examined as the first. Subsequently, the most suitable depth of the decision trees for sufficient generalization was scoped. Finally, the performance measures of models based on the proposed techniques were compared with the standardised reference TF-IDF technique. The performance measures of the model like accuracy, precision, recall and f1-score are considered, together with the 10-fold cross-validation technique. Simultaneously, the question, whether the TF-IDF technique can be improved using POS tags was researched in detail. The results showed that the newly proposed techniques are comparable with the traditional TF-IDF technique. At the same time, it can be stated that the morphological analysis can improve the baseline TF-IDF technique. As a result, the performance measures of the model, precision for fake news and recall for real news, were statistically significantly improved.

## INTRODUCTION

Fake news is currently the biggest bugbear of the developed world (*Jang et al., 2018*). Although the spreading of false information or false messages for personal or political benefit is certainly nothing new, current trends such as social media enable every individual to create false information easier than ever before (*Allcott & Gentzkow, 2017*).

The article deals with evaluating four proposed techniques for fake and true news classification using morphological analysis. Morphological analysis belongs to the basic

Corresponding author
Jozef Kapusta, jkapusta@ukf.sk

means for natural language processing research. It deals with the parts of speech tags (POS tags) as morphological characteristics of the word in the context, which can be considered a style-based fake detection method (*Zafarani et al., 2019*). Linguistic-based features are extracted from the text content in terms of document organisations from different levels, such as characters, words, sentences and documents. Sentence-level features refer to all the important attributes that are based on sentence scale. They include parts of speech tagging (POS), the average sentence length, the average length of a tweet/post, the frequency of punctuations, function words, and phrase in a sentence, the average polarity of the sentence (positive, neutral or negative), as well as the sentence complexity (*Khan et al., 2019*).

Existing research articles mainly investigate standard linguistic features, including lexical, syntactic, semantic and discourse features, to capture the intrinsic properties of misinformation. Syntactic features can be divided into shallow, where belongs frequency of POS tags and punctuations, and deep syntactic features (*Feng, Banerjee & Choi, 2012*). Morphological analysis of POS tags based on n-grams is used in this paper to evaluate its suitability for successful fake news classification.

An N-gram is a sequence of N tokens (words). N-grams are also called multi-word expressions or lexical bundles. N-grams can be generated on any attribute, with word and lemma being the most frequently used ones. The following word expressions represent 2-gram: 'New York', and 3-gram: 'The Three Musketeers'. The analysis of the n-grams is considered more meaningful than the analysis of the individual words (tokens), which constitute the n-grams.

Several research articles stated that simple content-related n-grams and POS tagging had been proven insufficient for the classification task (*Shu et al., 2017*; *Conroy, Rubin & Chen, 2015*; *Su et al., 2020*). However, these findings mainly represent the authors' opinion because they did not realise or publish any empirical research results, confirming these statements in the last decade.

Considering this contradiction, the main aim of the paper is to experimentally evaluate the potential of the common use of n-grams and POS tags for the correct classification of fake and true news. Therefore, continuous sequences of *n* items from a given sample of POS tags (n-grams) were analysed. The techniques based on POS tags were proposed and used in order to meet this aim. Subsequently, these techniques were compared with the standardised reference TF-IDF technique to evaluate their main performance characteristics. Simultaneously, the question of whether the TF-IDF technique can be improved using POS tags was researched in detail. All techniques have been applied in the pre-processing phase on different groups of n-grams. The resulted datasets have been analysed using decision tree classifiers.

The article aims to present and evaluate proposed techniques for pre-processing of input vectors of a selected classifier. These techniques are based on creating n-grams from POS tags. The research question is whether the proposed techniques are more suitable than the traditional baseline technique TF-IDF or whether these techniques are able to improve the results of the TF-IDF technique.

All proposed techniques have been applied to different levels of n-grams. Subsequently, the outcomes of these techniques were used as the input vectors of the decision tree classifier. The following methodology was used for evaluation of the suitability of a proposed approach based on n-grams of POS tags:

- Identification of POS tags in the analysed dataset.
- N-grams (1-grams, 2-grams, 3-grams, 4-grams) definition from POS tags. N-gram represents the sequence of the POS tags.
- Calculation of frequency of occurrence of an n-gram in documents. In other words, the relative frequency of n-gram in examined fake and true news is calculated.
- Definition of input vectors of classifiers using three proposed techniques for POS tags and controlled TF-IDF technique.
- Application of decision tree classifiers, parameter tuning concerning the different depths and length of n-grams.
- Identification and comparison of the decision trees' characteristics, mainly the accuracy, depth of the trees and time performance.

The structure of the article is as follows. The current state of the research in the field of fake news identification is summarised in the second section. The datasets of news Covid-19 used in the research are described in the second section. This section also describes the process of n-grams extraction from POS tags. Simultaneously, three POS tags-based techniques are proposed for preparing input vectors for decision trees classifiers. Subsequently, the same section discusses the process of decision trees modelling, the importance of finding the most suitable n-gram length and maximum depth. Finally, statistical evaluation of the performance of the modified techniques based on POS tags for fake news classification is explained in the same section. The most important results, together with an evaluation of model performance and time efficiency of the proposed techniques, are summarised in the fourth section. The detailed discussion about the obtained results and conclusions form the content of the last section of the article.

## RELATED WORK

There has been no universal definition for fake news. However, *Zhou & Zafarani (2020)* define fake news as intentionally false news published by a news outlet. Simultaneously, they explained related terms in detail and tried to define them with a discussion about the differences based on the huge set of related publications. The same authors categorised automatic detection of fake news from four perspectives: knowledge, style, propagation and source. Considering this, the research described in this paper belongs to the style-based fake news detection category, which methods try to assess news intention (*Zhou & Zafarani, 2020*). According to their definition, fake news style can be defined as a set of quantifiable characteristics (features) that can well represent fake news content and differentiate it from true news content.

*Kumar & Shah (2018)* provided a comprehensive review of many facets of fake news distributed over the Internet. They quantified the impact of fake news and characterised

the algorithms used to detect and predict them. Moreover, they summarise the current state of the research and approaches applied in the field of fake news content analysis from the linguistic, semantic and knowledge discovery point of view. They did not conclude the overall performance of the style-based methods using ML algorithms despite the overall scope of the review.

Other contemporary surveys (*Zhou & Zafarani, 2020*; *Zhang & Ghorbani, 2020*; *Shu et al., 2017*) provide further evidence that the research related to the field of fake news is very intense now, mainly due to their negative consequences for society. The authors analysed various aspects of the fake news research, discussed the reasons, creators, resources and methods of their dissemination, as well as the impact and the machine learning algorithms created to detect them effectively.

*Sharma et al. (2019)* also published a comprehensive survey highlighting the technical challenges of fake news. They summarised characteristic features of the datasets of news and outlined the directions for future research. They discussed existing methods and ML techniques applicable to identifying and mitigating fake news, focusing on the significant advances in each method and their advantages and limitations. They discussed the results of the application of different classification algorithms, including decision trees. They concluded that using n-grams alone can not entirely capture finer-grained linguistic information present in fake news writing style. However, their application on the dataset, which contains pre-processed items using POS tagging, is not mentioned.

*Zhang & Ghorbani (2020)* stated that because online fake reviews and rumours are always compacted and information-intensive, their content lengths are often shorter than online fake news. As a result, traditional linguistic processing and embedding techniques such as bag-of-words or n-gram are suitable for processing reviews or rumours. However, they are not powerful enough for extracting the underlying relationship for fake news. For online fake news detection, sophisticated embedding approaches are necessary to capture the key opinion and sequential semantic order in news content.

*de Oliveira et al. (2021)* realized the literature survey focused on the preprocessing data techniques used in natural language processing, vectorization, dimensionality reduction, machine learning, and quality assessment of information retrieval. They discuss the role of n-grams and POS tags only partially.

On the other hand, *Li et al. (2020)* consider the n-gram approach the most effective linguistic analysis method applied to fake news detection. Apart from word-based features such as n-grams, syntactic features such as POS tags are also exploited to capture linguistic characteristics of texts.

*Stoick, Snell & Straub (2019)* stated that previous linguistic work suggests part-of-speech and n-gram frequencies are often different between fake and real articles. He created two models and concluded that some aspects of the fake articles remained readily identifiable, even when the classifier was trained on a limited number of examples. The second model used n-gram frequencies and neural networks, which were trained on n-grams of different length. He stated that the accuracy was near the same for each n-gram size, which means that some of the same information may be ascertainable across n-grams of different sizes. *Ahmed, Traore & Saad (2017)* further argued that the latest advance

in natural language processing (NLP) and deception detection could help to detect deceptive news. They proposed a fake news detection model that analyses n-grams using different features extraction and ML classification techniques. The combination of TF-IDF as features extraction, together with LSVM classifier, achieved the highest accuracy. Similarly, *Zhou et al. (2020)* extracted linguistic/stylometric features, a bag of words TF and BOW TF-IDF vector and applied the various machine learning models, including bagging and boosting methods, to achieve the best accuracy. However, they stated that the lack of available corpora for predictive modelling is an essential limiting factor in designing effective models to detect fake news.

*Wynne & Wint (2019)* investigated two machine learning algorithms using word n-grams and character n-grams analysis. They obtained better results using character n-grams with TF-IDF and Gradient Boosting Classifier. They did not discuss the pre-processing phase of n-grams, as will be described in this article.

*Thorne & Vlachos (2018)* surveyed automated fact-checking research stemming from natural language processing and related disciplines, unifying the task formulations and methodologies across papers and authors. They identified the subject-predicate-object triples from small knowledge graphs to fact check numerical claims. Once the relevant triple had been found, a truth label was computed through a rule-based approach that considered the error between the claimed values and the retrieved values from the graph.

*Shu et al. (2017)* proposed to use linguistic-based features such as total words, characters per word, frequencies of large words, frequencies of phrases (i.e., n-grams and bag-of-words). They stated that fake contents are generated intentionally by malicious online users, so it is challenging to distinguish between fake information and truth information only by content and linguistic analysis.

POS tags were also exploited to capture the linguistic characteristics of the texts. However, several works have found the frequency distribution of POS tags to be closely linked to the genre of the text being considered (*Sharma et al., 2019*).

*Ott et al. (2011)* examined this variation in POS tag distribution in spam, intending to find if this distribution also exists concerning text veracity. They obtained better classification performance with the n-grams approach but found that the POS tags approach is a strong baseline outperforming the best human judge. Later work has considered more in-depth syntactic features derived from probabilistic context-free grammars (PCFG) trees. They assumed that the approach based only on n-grams is simple and cannot model more complex contextual dependencies in the text. Moreover, syntactic features used alone are less powerful than word-based n-grams, and a naive combination of the two cannot capture their complex interdependence. They concluded that the weights learned by the classifier are mainly in agreement with the findings of existing theories on deceptive writing (*Ott, Cardie & Hancock, 2013*).

Some authors, for example, *Conroy, Rubin & Chen (2015)*, have noted that simple content-related n-grams and POS tagging have been proven insufficient for the classification task. However, they did not research the n-grams from the POS tags. They suggested using Deep Syntax analysis using Probabilistic Con-text-Free Grammars

(PCFG) to distinguish rule categories (lexicalised, non-lexicalised, parent nodes, etc.) instead of deception detection with 85–91% accuracy.

*Su et al. (2020)* also stated that simple content-related n-grams and shallow part-of-speech (POS) tagging have proven insufficient for the detection task, often failing to account for important context information. On the other hand, these methods have been proven useful only when combined with more complex analysis methods.

*Khan et al. (2019)* stated that meanwhile, the linguistic-based features extracted from the news content are not sufficient for revealing the in-depth underlying distribution patterns of fake news (*Shu et al., 2017*). Auxiliary features, such as the news author's credibility and the spreading patterns of the news, play more important roles for online fake news prediction.

On the other hand, *Qian et al. (2018)* proposed a similar approach, which is researched further in this paper, based on a convolutional neural network (TCNN) with a user response generator (URG). TCNN captures semantic information from text by representing it at the sentence and word level. URG learns a generative user response model to a text from historical user responses to generate responses to new articles to assist fake news detection. They used POS tags in combination with n-grams as a comparison of the accuracy of the proposed technique of NN based classification.

*Goldani, Momtazi & Safabakhsh (2021)* used capsule neural networks in the fake news detection task. They applied different levels of n-grams for feature extraction and subsequently used different embedding models for news items of different lengths. Static word embedding was used for short news items, whereas non-static word embeddings that allow incremental uptraining and updating in the training phase are used for medium length or long news statements. They did not consider POS tags in the pre-processing phase.

Finally, *Kapusta et al. (2020)* realised a morphological analysis of several news datasets. They analysed the morphological tags and compared the differences in their use in fake news and real news articles. They used morphological analysis for words classification into grammatical classes. Each word was assigned a morphological tag, and these tags were thoughtfully analysed. The first step consisted of creating groups that consisted of related morphological tags. The groups reflected on the basic word classes. The authors identified statistically significant differences in the use of word classes. Significant differences were identified for groups of foreign words, adjectives and nouns favouring fake news and groups of wh-words, determiners, prepositions, and verbs favouring real news. The third dataset was evaluated separately and was used for verification. As a result, significant differences for groups adverb, verbs, nouns were identified. They concluded that it is important that the differences between groups of words exist. It is evident that morphological tags can be used as input into the fake news classifiers.

## MATERIALS & METHODS

### Dataset

The dataset analysed by *Li (2020)* was used for the evaluation of proposed techniques. This dataset collects more than 1,100 articles (news) and posts from social networks related

**Table 1 Morphological tags used for news classification (*Schmid, 1994*).**

| GTAG | POS Tags |
| --- | --- |
| group C | CC (coordinating conjunction), CD (cardinal number) |
| group D | DT (determiner) |
| group E | EX (existential there) |
| group F | FW (foreign word) |
| group I | IN (preposition, subordinating conjunction) |
| group J | JJ (adjective), JJR (adjective, comparative), JJS (adjective, superlative) |
| group M | MD (modal) |
| group N | NN (noun, singular or mass), NNS (noun plural), NNP (proper noun, singular), NNPS (proper noun, plural), |
| group P | PDT (predeterminer), POS (possessive ending), PP (personal pronoun) |
| group R | RB (adverb), RBR (adverb, comparative), RBS (adverb, superlative), RP (particle) |
| group T | TO (infinitive 'to') |
| group U | UH (interjection) |
| group V | VB (verb be, base form), VBD (verb be, past tense), VBG (verb be, gerund/present participle), VBN (verb be, past participle), VBP (verb be, sing. present, non-3d), VBZ (verb be, 3rd person sing. present) |
| group W | WDT (wh-determiner), WP (wh-pronoun), WP$ (possessive wh-pronoun), WRB (wh-abverb) |

to Covid-19. It was created in cooperation with the projects Lead Stories, Poynter, FactCheck.org, Snopes, EuVsDisinfo, which monitor, identify and control misleading information. These projects define the true news as an article or post, which truthfulness can be proven and come from trusted resources. Vice versa, as the fake news are considered all articles and post, which have been evaluated as false and come from known fake news resources trying to broadcast misleading information intentionally.

## POS tags

Morphological tags were assigned to all words of the news from the dataset using the unique tool called TreeTagger. *Schmid (1994)* developed the set of tags called English Penn Treebank using this annotating tool. The final English Penn Treebank tagset contains 35 morphological tags. However, considering the aim of the research, the following tags were not included in the further analysis due to their low frequency of appearance or discrepancy:

- SYM (symbol),
- LS (list marker).

Therefore, the final number of morphological tags used in the analysis was 33. Table 1 shows the morphological tags divided into groups.

## N-grams extraction from POS tags

N-grams were extracted from POS tags in this data pre-processing step. As a result, sequences of n-grams from a given sample of POS tags were created. Figure 1 demonstrates this process using the sentence from the tenth most viewed fake news story shared on
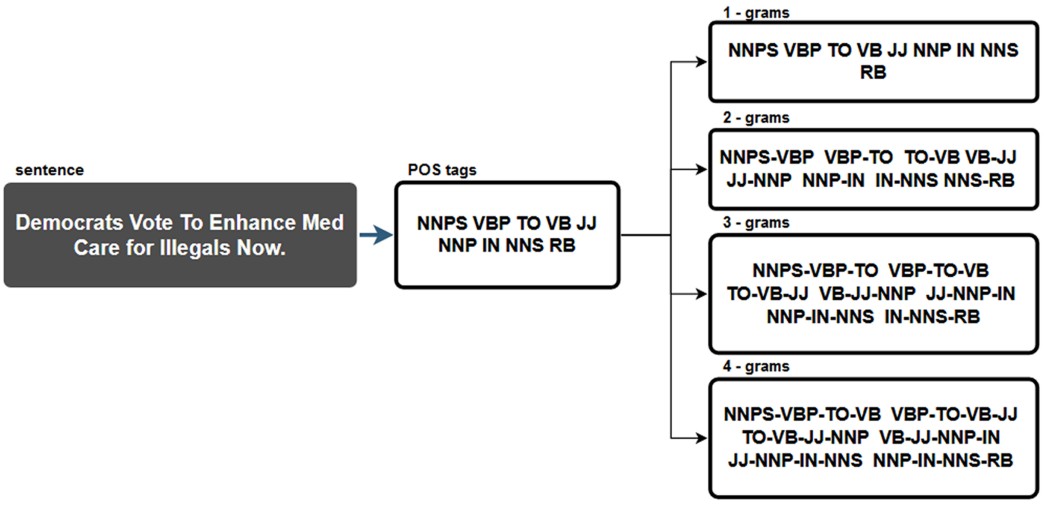

**Figure 1  Example of n-gram extraction from one sentence.**

Facebook in 2019. The following POS tags were identified from the sentence "Democrats Vote To Enhance Med Care for Illegals Now":

- NNPS (proper noun, plural),
- VBP (verb, sing. present, non-3d),
- TO (to),
- VB (verb, base form),
- JJ (adjective),
- NNP (proper noun, singular),
- IN (preposition/subordinating conjunction),
- NNS (noun plural),
- RB (adverb).

Since 1-grams and identified POS tags are identical, the input file with 1-grams used in further research is identical to the file with identified POS tags. The n-grams for the TF-IDF technique were created in the same way. However, it is important to emphasise that this technique used so call terms, which represent the lemmas or stems of a word.

## The techniques used to pre-process the input vectors

The following four techniques have been applied for pre-processing of the input vectors for a selected classifier.

### Term frequency - inverse document frequency (TF-IDF) technique

TF-IDF is a traditional technique that leveraged to assess the importance of tokens to one of the documents in a corpus (*Qin, Xu & Guo, 2016*). The TF-IDF approach creates a bias in that frequent terms highly related to a specific domain, which is typically identified as noise, thus leading to the development of lower term weights because the traditional

TF-IDF technique is not specifically designed to address large news corpora. Typically, the TF-IDF weight is composed of two terms: the first computes the normalised Term Frequency (TF), the second term is the Inverse Document Frequency (IDF).

Let $t$ is a term/word, $d$ is a document, $w$ is any term in the document. Then the frequency of the term/word $t$ in document $d$ is calculated as follows

$$tf(t,d) = \frac{f(t,d)}{f(w,d)},$$

where $f(t,d)$ is the number of terms/words in document $d$ and $f(w,d)$ is the number of all terms in the document. Simultaneously, the number of all documents is also taken into account in TF-IDF calculation, in which a particular term/word occurs. This number is denoted as $idf(t,D)$. It represents an inverse document frequency expressed as follows

$$idf(t,D) = \ln \frac{N}{\sum(d \in D \; : \; t \in d) + 1},$$

where $D$ is a corpus of all documents and $N$ is a number of documents in the corpus.

The formula of TfIdf can be written as

$$tfidf(t,d,D) = tf(t,d) \; \times \; idf(t,D).$$

Formula $tf$ has various variants such as $\log(tf(t,d))$ or $\log(tf(t,d) + 1)$. Similarly, there are several variants, how $idf$ can be calculated (*Chen, 2017*). Considering this fact, the calculation of the TfIdf was realised using the scikit-learn library in Python (https://scikit-learn.org). The TF-IDF technique applied in the following experiment is used as a reference technique for comparison selected characteristics of the new techniques described below. The same dataset was used as an input. However, the stop words were removed before in this case.

### POS frequency (PosF) technique

This technique is an analogy of the Term Frequency technique. However, it calculates with the frequency of POS tags.

Let $pos$ is an identified POS tag, $d$ document, $w$ represents any POS tag identified in the document. Then the frequency of POS tag $pos$ in document $d$ can be calculated as follows:

$$PosF(pos,d) = \frac{f(pos,d)}{f(w,d)},$$

where $f(pos,d)$ is the number of occurrences of POS tag in document $d$ and $f(w,d)$ is the number of all identified POS tags in the document.

As a result, PosF expresses the relative frequency of each POS tag in the frame of the analysed list of POS tags identified in the document.

### PosF-IDF technique

This technique is the analogy of the TF-IDF technique. Similarly to the already introduced PosF technique, it considers the POS tags, which have been identified in each document

in the analysed dataset based on individual words and sentences. The documents containing only identified POS tags represented the inputs for the calculation of PosF-IDF. Besides the relative frequency of POS tags in the document, the number of all documents in which a particular POS tag has also been identified is considered.

### Merged TF-IDF and PosF technique

This technique was proposed to confirm whether is it possible to improve the traditional TF-IDF technique by using POS tags. Therefore, the following vectors were created for each document:

- TfIdf vector,
- PosF vector, which represents the relative frequency of POS tags in the document.

Subsequently, a result of applying the merged technique is again a vector, which originated by merging the previous vectors. Therefore, both vectors $\overrightarrow{TfIdf(d)}$ a $\overrightarrow{PosF(d)}$ are considered for document d, which were calculated using the techniques TfIdf and PosF mentioned.

$$\overrightarrow{TfIdf(d)} = (t_1, t_2, \ldots, t_n),$$

$$\overrightarrow{PosF(d)} = (p_1, p_2, \ldots, p_m).$$

Then, the final vector $\overrightarrow{merge(d)}$ for document $d$ calculated by the merge technique is

$$\overrightarrow{merge(d)} = (t_1, t_2, \ldots, t_m, p_1, p_2, \ldots, p_m).$$

A set of techniques for pre-processing the input vectors for the selected knowledge discovery classification task was created. These techniques can be considered the variations of the previous TF-IDF technique, in which the POS tags are taken into account additionally to the original terms.

As a result, the four techniques described above represent typical variations, which allow comparing and analysing the basic features of the techniques based on the terms and POS tags.

## Decision trees modelling

Several classifiers like decision tree classifiers, Bayesian classifiers, k-nearest-neighbour classifiers, case-based reasoning, genetic algorithms, rough sets, and fuzzy logic techniques were considered. Finally, the decision trees were selected to evaluate the suitability of the proposed techniques for calculating the input vectors and analyse their features. The decision trees allow not only a simple classification of cases, but they create easily interpretable and understandable classification rules at the same time. In other words, they simultaneously represent functional classifiers and a tool for knowledge discovery and understanding. The same approach was partially used in other similar research papers (*Kapusta et al., 2020*; *Kapusta, Benko & Munk, 2020*).

The attribute selection measures like Information Gain, Gain Ratio, and Gini Index (*Lubinsky, 1995*), used while decision tree is created, are considered the further important

factor, why decision trees had been finally selected. The best feature is always selected in each step of decision tree development. Moreover, it is virtually independent of the number of input attributes. It means that even though there is supplemented a larger amount of the attributes (elements of the input vector) on the input of the selected classifier, the accuracy remains unchanged.

## K-fold validation

Comparing the decision trees created in the realised experiment is based on the essential characteristics of the decision trees as the number of nodes or leaves. These characteristics define the size of the tree, which should be suitably minimised. Simultaneously, the performance measures of the model like accuracy, precision, recall and f1-score are considered, together with 10-fold cross-validation technique.

K-fold validation was used for the evaluation of the models. It generally results in a less biased model compare to other methods because it ensures that every observation from the original dataset has the chance of appearing in training and test set.

## Setting the most suitable n-gram length

All compared techniques for input vectors pre-processing required identical conditions. Therefore, the highest values of n in n-grams was determined as the first step. Most NLP tasks work usually with $n = \{1,2,3\}$. The higher value of n (4-grams, 5-grams, etc.) has significant demands on hardware and software, calculation time, and overall performance. On the other hand, the potential contribution of the higher n-grams in increasing the accuracy of created models is limited.

Several decision tree models were created to evaluate this consideration. N-grams (1-gram, 2-gram, …, 5-gram) for tokens/words and for POS tags were prepared. Subsequently, the TF-IDF technique was applied to n-grams of tokens/words. At the same time, PosF and PosfIdf techniques were applied on n-grams of POS tags. As a result, 15 files with the input vectors have been created (1–5-grams × 3 techniques). Figure 2 visualises the individual steps of this process for better clarity.

Ten-fold cross-validation led to creating ten decision trees models for each pre-processed file (together 15 files). In all cases, the accuracy was considered the measure of the model performance. Figure 3 shows a visualisation of all models with different n-gram length. The values on the x-axis represent a range of used n-grams. For instance, n-gram (1,1) means that only unigrams had been used. Other ranges of n-grams will be used in the next experiments. For example, the designation of (1,4) will represent the 1-grams, 2-grams, 3-grams, and 4-grams included together in one input file in this case.

The results show that the accuracy is declining with the length of the n-grams, mostly in the case of applying the TF-IDF technique. Although it was not possible to process longer n-grams (6-grams, 7-grams, etc.) due to the limited time and computational complexity, it can be assumed that their accuracy would be declined similarly to the behaviour of the accuracy for 5-grams in the case of all applied techniques.

Considering the process of decision tree model creation, it is not surprising that joining the n-grams to one input file achieved the highest accuracy. The best accuracy can be

**Peer**J Computer Science

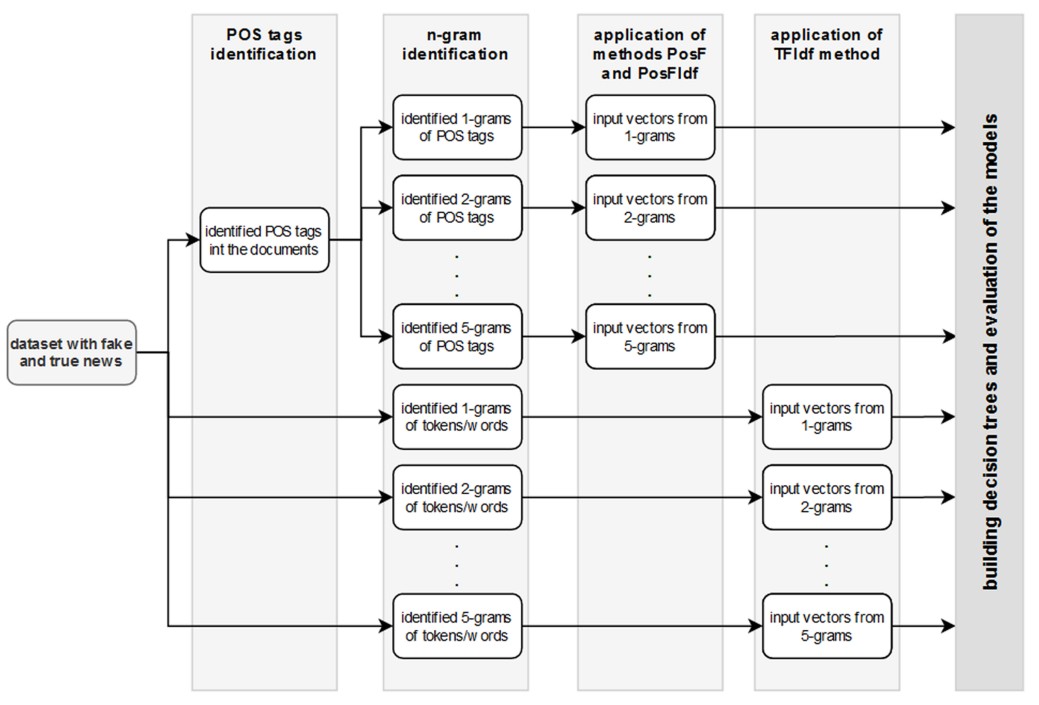

**Figure 2** The workflow of the experiment.

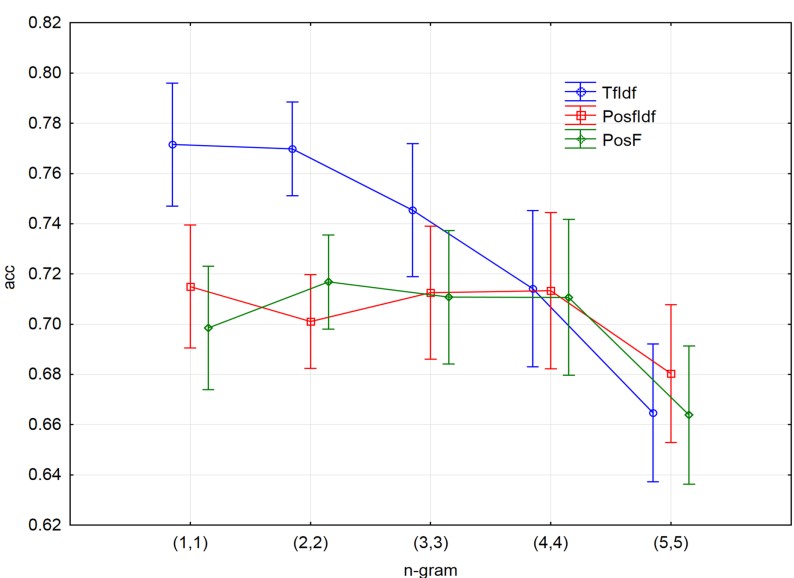

**Figure 3** The results of 10-fold cross-validation accuracy for different n-grams and three techniques of input vectors pre-processing.

reached by joining n-grams to one input file. Considering this, the most suitable measure will be selected during the creation of the decision tree. As a result, all following experiments will work with the file, consisting of joined 1-grams, 2-grams, 3-grams and 4-grams (1,4).

**Table 2 Selected characteristics of the complete decision trees.**

|  | TfIdf(1,4) | PosfIdf(1,4) | PosF(1,4) |
|---|---|---|---|
| avarage (deep) | 25.1 | 15.6 | 17.2 |
| min (deep) | 22 | 12 | 11 |
| max (deep) | 30 | 19 | 21 |
| avarage (node count) | 171.4 | 157.4 | 156 |
| avarage (leaf count) | 86.2 | 79.2 | 78.5 |
| average (number of vectors elements) | 817,213.3 | 69,512.4 | 69,775 |

## Setting the maximum depth of the decision tree

Overfitting represents a frequent issue. Although the training error decreases by default with the increasing size of the created tree, the test errors often increase with the increasing size. As a result, the classification of new cases can be inaccurate. Techniques like pruning or hyperparameter tuning can overcome overfitting.

The maximal depth of the decision tree will be analysed to minimise the overfitting issue to find understandable rules for fake news identification.

As was mentioned earlier, the main aim of the article is to evaluate the most suitable techniques for the preparation of input vectors. Simultaneously, the suitable setting of the parameter max_depth will be evaluated. Complete decision trees for n-grams from tokens/words and POS tags were created for finding suitable values of selected characteristics of decision trees (Table 2).

The results show that the techniques working with the POS tags have a small number of input vectors compared to the reference TF-IDF technique. These findings were expected because while TF-IDF takes all tokens/words, in the case of the PosfIdf as well as PosF techniques, each token/word had been assigned to one of 33 POS tags (Table 1). This simplification is also visible in the size of the generated decision tree (depth, node count, number of leaves). The application of the PosfIdf and PosF techniques led to the simpler decision tree.

However, the maximal depth of the decision tree is the essential characteristics for further considerations. While it is equal to 30 for TF-IDF, the maximal depth is lower in the case of both remaining techniques. Therefore, decision trees with different depths will be further considered in the main experiment to ensure the same conditions for all compared techniques. The maximal depth will be set to 30.

## The methodology of the main experiment

The following experiment's main aim is to evaluate if it is possible to classify the fake news messages using POS tags and compare the performance of the proposed techniques (PosfIdf, PosF, merge) with the reference TF-IDF technique which uses tokens/words.

The comparison of these four techniques is joined with the following questions:

Q1: What is the most suitable length of the n-grams for these techniques?

Q2: How to create models using these techniques to prevent possible overfitting?

Q3: How to compare the models with different hyperparameters, which tune the performance of the models?

The first question (Q1) was answered in the section Setting the Most Suitable N-gram Length. As its result, the use of joined 1-grams, 2-grams, 3-grams and 4-grams (1,4) is the most suitable. The second question (Q2) can be answered by experimenting with the maximum depth hyperparameter used in the decision tree classifier. The highest acceptable value of this hyperparameter was found in the section Setting the Maximum Depth of the Decision Tree. The main experiment described later will be realised regarding the last, third question (Q3). The following steps of the methodology will be applied:

1. Identification of POS tags in the dataset.
2. Application of PosF and PosfIdf input vector preparation techniques on identified POS tags.
3. Application of reference TF-IDF technique to create input vectors. This technique uses tokens to represent the words modified by the stemming algorithm. Simultaneously, the stop words are removed.
4. Joining PosF and TF-IDF technique to merge vector.
5. Iteration with different values of maximal depth (1, …, 30):

   - Randomised distribution of the input vectors of PosF, PosfIdf, TfIdf, and Merge techniques into training and testing subsets in accordance with the requirements of the 10-fold cross-validation.

   - Calculation of decision tree for each training subset with the given maximal depth.
   - Testing the quality of the model's predictions on the testing subset. The following new characteristics were established:
   - *prec_fake* (precision for group fake),
   - *prec_real* (precision for group real),
   - *rec_fake* (recall for group fake),
   - *rec_real* (recall for group real),
   - *f1-score*,
   - *time* spent on one iteration.
   - Analysis of the results (evaluation of the models).

The results of steps 1–4 are four input vectors prepared using the before-mentioned four proposed techniques. The fifth step of the proposed methodology is focused on the evaluation of these four examined techniques.

The application of the proposed methodology with 10-fold cross-validation resulted in the creation of 1,200 different decision trees (30 max_depth values × 4 techniques × 10-fold validation). In other words, 40 decision trees with 10-fold cross-validation were created for each maximal depth. Figure 4 depicts the individual steps of the methodology of the experiment.

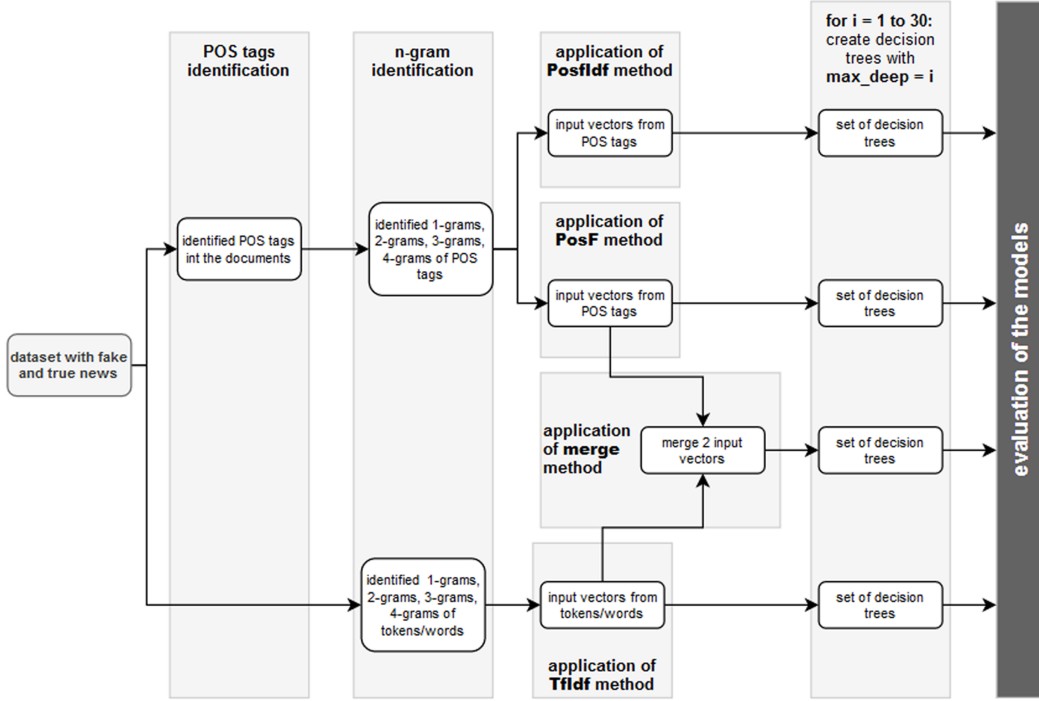

**Figure 4** The individual steps of the experiment for comparison of 4 proposed techniques, 30 values of maximal depth and 10-fold cross-validation.

The last step of the proposed methodology, analysis of the result, will be described in section Results. All steps of the methodology were implemented in Python and its libraries. Text processing was realised using the NLTK library (https://www.nltk.org/). The tool TreeTagger (*Schmid, 1994*) was used for the identification of POS tags. Finally, the scikit-learn library (https://scikit-learn.org) was used for creating decision tree models.

The Gini impurity function was applied to measure the quality of a split of decision trees.

The strategy used to choose the split at each node was chosen "best" split (an alternative is "best random split"). Subsequently, the maximum depth of the decision trees was examined to prevent overfitting. Other hyperparameters besides the minimum number of samples required to split an internal node or the minimum number of samples required to be at a leaf node were not applied.

# RESULTS

## Experiment

The quality of the proposed models (TfIdf(1,4), PosfIdf(1,4), PosF(1,4), merge(1,4)) was evaluated using evaluation measures (prec, rec, f1-sc, prec_fake, rec_fake, prec_real, rec_real), as well as from time effectivity point of view (*time*). A comparison of the depths of the complete decision trees showed (Table 2) that there is no point in thinking about the depth greater than 29. Therefore, decision trees with a maximal depth less than 30 were created in line with the methodology referred to in "Setting the Maximum Depth of the Decision Tree".

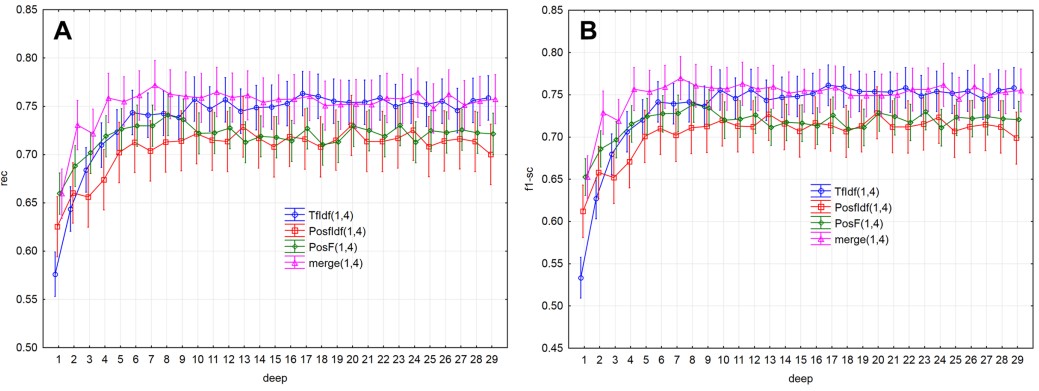

**Figure 5 Point and interval mean estimation for the model performance measures: (A) rec, (B) f1- sc.**

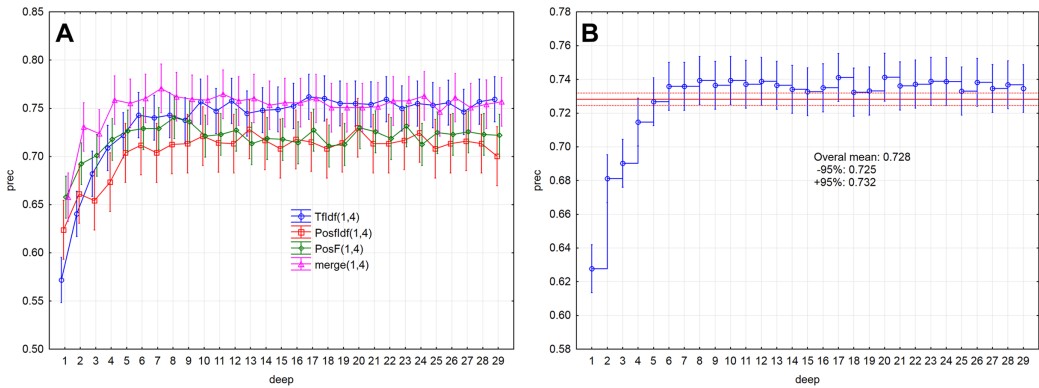

**Figure 6 Point and interval estimation of means for the model performance measure prec: (A) model, (B) total.**

Evaluation measures (Figs. 5A, 5B and 6A ) increase up to the depth of five and reach the values smaller than 0.73 in average (rec < 0.727, f1-sc < 0.725, prec < 0.727). Subsequently, they reach stable values greater than 0.73 from the depth of six (rec > 0.732, f1-sc > 0.731, prec > 0.732) and less than 0.75 (rec < 0.742, f1-sc < 0.740, prec < 0.741)

As a result, the PosF technique reaches better performance in small values of depth (up to 4) compared to others. While the merge technique originates from the joining of PoSF and TF-IDF technique, its results will be naturally better.

The model performance (prec) for the given depths (<30) reached the above-average values from the depth of six (Fig. 2B). The model performance measure prec ($p > 0.05$) was not statistically significant differences from depth equal to six. Similar results were also obtained for measures rec and f1-sc. As a result, the models' performance will be further examined for depths 6–10.

The Kolmogorov–Smirnov test was applied to verify the normality assumption. The examined variables (*model × evaluation measure, model × time*) have normal distribution for all levels of the between-groups factor *deep* (6: *max D* < 0.326, $p > 0.05$, 7: *max D* < 0.247, $p > 0.05$, 8: *max D* < 0.230, $p > 0.05$, 9: *max D* < 0.298, $p > 0.05$, 10: *max D* < 0.265, $p > 0.05$).

The Mauchley sphericity test was consequently applied for verifying the covariance matrix sphericity assumption for repeated measures with four levels (TfIdf(1,4), PosfIdf (1,4), PosF(1,4), merge(1,4)) with the following results (prec: $W = 0.372$, *Chi-Square* = 43.217, $p < 0.001$; rec: $W = 0.374$, *Chi-Square* = 43.035, $p < 0.001$; f1-sc: $W = 0.375$, *Chi-Square* = 42.885, $p < 0.001$; prec_fake: $W = 0.594$, *Chi-Square* = 22.809, $p < 0.001$; rec_fake: $W = 0.643$, *Chi-Square* = 19.329, $p < 0.01$; prec_real: $W = 0.377$, *Chi-Square* = 42.641, $p < 0.001$; rec_real: $W = 0.716$, *Chi-Square* = 14.599, $p < 0.05$; time: $W = 0.0001$, *Chi-Square* = 413.502, $p < 0.001$).

This test was statistically significant in all cases of examined evaluation measures and time ($p < 0.05$). It means that the assumption was violated because unless the assumption of covariance matrix sphericity is not met, the I. type error increases (*Ahmad, 2013*; *Haverkamp & Beauducel, 2017*; *Munkova et al., 2020*).

Therefore, the degrees of freedom had been adjusted $df1 = (J - 1)(I - 1)$, $df2 = (N - l)(I - 1)$ for the used *F*-test using Greenhouse-Geisser and Huynh-Feldt adjustments (*Epsilon*). As a result, the declared level of significance was reached

$$adj.df1 = \widehat{Epsilon}(J - 1)(I - 1),$$
$$adj.df2 = \widehat{Epsilon}(N - l)(I - 1),$$

where $I$ is the number of levels of the factor *model* (dependent samples), $J$ is the number of the levels of the factor *deep* (independent samples), and $N$ is the number of cases.

The Bonferroni adjustment was used to apply multiple comparisons. This adjustment is usually applied when several dependent and independent samples are simultaneously compared (*Lee & Lee, 2018*; *Genç & Soysal, 2018*). Bonferroni adjustment represents the most conservative approach, in which the level of significance (*alpha*) for a whole set $N$ of cases is set so that the level of significance for each case is equal to $\frac{alpha}{N}$.

## Model performance

The first phase of the analysis focused on the performance of the models. The performance was analysed by selected evaluation measures (prec, rec, f1-sc, prec_fake, rec_fake, prec_real, rec_real) according to the within-group factor and between-groups factor and their interaction. The models (TfIdf(1,4), PosfIdf(1,4), PosF(1,4), merge(1,4)) represented the levels of within-group factor. The depths of the decision tree (6-10) represented the levels of a between-group factor. Considering the violated assumption of covariance matrix sphericity, the modified tests for repeated measures were applied to assess the effectivity of the examined models (*Dien, 2017*; *Montoya, 2019*). *Epsilon* represented the degree of violation of this assumption. If *Epsilon* equals one, the assumption is fulfilled. The values of *Epsilon* were significantly lower than one in both cases (*Epsilon* $< 0.69$). Zero hypotheses claim that there is no statistically significant difference in the quality of the examined models. The zero hypotheses, which claimed that there is not a statistically significant difference in values of evaluation measures prec, rec, and f1-sc between examined models, were rejected at the 0.001 significance level (prec: *G-G Epsilon* = 0.597, *H-F Epsilon* = 0.675, *adj.p* $< 0.001$; rec: *G-G Epsilon* = 0.604, *H-F Epsilon* = 0.684, *adj.p* $< 0.001$; f1-sc: *G-G Epsilon* = 0.599, *H-F Epsilon* = 0.678, *adj.p* $< 0.001$). On the contrary, the zero hypotheses,

**Table 3 Bonferroni adjustment for the model performance measures: (A) prec, (B) rec, (C) f1-sc.**

| Model | prec Mean | 1 | 2 | 3 | rec Mean | 1 | 2 | 3 | f1-sc Mean | 1 | 2 | 3 |
|---|---|---|---|---|---|---|---|---|---|---|---|---|
| PosfIdf(1,4) | 0.712 | **** | | | 0.713 | **** | | | 0.711 | **** | | |
| PosF(1,4) | 0.731 | **** | **** | | 0.732 | **** | **** | | 0.730 | **** | **** | |
| TfIdf(1,4) | 0.744 | | **** | **** | 0.745 | | **** | **** | 0.743 | | **** | **** |
| merge(1,4) | 0.762 | | | **** | 0.763 | | | **** | 0.761 | | | **** |

Note:
**** Homogeneous groups ($p > 0.05$).

**Table 4 Bonferroni adjustment for the model performance measures: (A) prec_fake, (B) rec_fake.**

| Model | prec_fake Mean | 1 | 2 | 3 | Model | rec_fake Mean | 1 | 2 |
|---|---|---|---|---|---|---|---|---|
| PosfIdf(1,4) | 0.705 | **** | | | PosfIdf(1,4) | 0.725 | **** | |
| PosF(1,4) | 0.717 | **** | **** | | merge(1,4) | 0.752 | **** | **** |
| TfIdf(1,4) | 0.738 | | **** | | TfIdf(1,4) | 0.754 | **** | **** |
| merge(1,4) | 0.768 | | | **** | PosF(1,4) | 0.760 | | **** |

Note:
**** Homogeneous groups ($p > 0.05$).

which claimed that the performance of the models (prec/rec/f1-sc) does not depend on a combination of within-group factor and between-groups factor, were not rejected ($p > 0.05$) (*model × deep*). Factor *deep* has not any impact on the performance of the examined models.

After rejecting the global zero hypotheses, the statistically significant differences between the models in the quality of the model's predictions were researched. Three homogeneous groups were identified based on prec, rec and f1-sc using the multiple comparisons. PosF(1,4) and TfIdf(1,4) techniques reached the same quality of the model's predictions ($p > 0.05$). Similar results were obtained for the pair PosfIdf(1,4) and PosF(1,4), as well for the pair TfIdf(1,4) and merge(1,4). Statistically significant differences in the quality of the model's predictions (Table 3) were identified between the models merge(1,4) and Pos ($p < 0.05$), as well as between the models TfIdf(1,4) and PosfIdf(1,4) ($p < 0.05$). The merge(1,4) model reached the highest quality, considering the evaluation measures.

The values of *Epsilon* were smaller than one in the case of partial evaluation measures prec_fake and rec_fake for the fake news. This finding was more notable in the case of Greenhouse–Geisser correction (*Epsilon* < 0.78). The zero hypotheses, which claimed that there is not any significant difference between the values of evaluation measures prec_fake and rec_fake between the examined models, were rejected (prec_fake: *G-G Epsilon* = 0.779, *H-F Epsilon* = 0.897, *adj.p* < 0.001; rec_fake: *G-G Epsilon* = 0.756, *H-F Epsilon* = 0.869, *adj.p* < 0.001). The impact of the between-groups factor *deep* has not been proven ($p > 0.05$). The performance of the models (prec_fake/rec_fake) does not depend on the interaction of the factors *model* and *deep*.

Two homogeneous groups were identified for prec_fake (Table 4A). PosF(1,4) and TfIdf (1,4), as well as PosF(1,4) and PosfIdf(1,4) reached the same quality of the model's predictions ($p > 0.05$). The statistically significant differences in the quality of the model's predictions (Table 4A) were identified between merge(1,4) and other models ($p < 0.05$) and

**Table 5 Bonferroni adjustment for the model performance: (A) prec_real, (B) rec_real.**

| Model | prec_real Mean | 1 | 2 | Model | rec_real Mean | 1 | 2 | 3 |
|---|---|---|---|---|---|---|---|---|
| PosfIdf(1,4) | 0.721 | | **** | PosfIdf(1,4) | 0.701 | **** | | |
| PosF(1,4) | 0.749 | **** | | PosF(1,4) | 0.704 | **** | **** | |
| TfIdf(1,4) | 0.752 | **** | | TfIdf(1,4) | 0.735 | | **** | |
| merge(1,4) | 0.761 | **** | | merge(1,4) | 0.774 | | | **** |

Note:
**** Homogeneous groups ($p > 0.05$).

between TfIdf(1,4) and PosfIdf(1,4) ($p < 0.05$). The merge(1,4) model reached the best quality from the prec_fake point of view.

On the other hand, PosF(1,4) model reached the highest quality considering the results of the multiple comparison for rec_fake (Table 4B). Two homogeneous groups (PosfIdf (1,4), merge(1,4), TfIdf(1,4)) and (merge(1,4), TfIdf(1,4), PosF(1,4)) were identified based on the evaluation measure rec_fake (Table 3B). The statistically significant differences (Table 3B) were identified only between the PosF(1,4) and PosfIdf(1,4) models ($p < 0.05$).

Similarly, the values of *Epsilon* were smaller than one (*G-G Epsilon* < 0.85) in case of evaluation measures prec_real and rec_real, which evaluate the quality of the prediction for a partial class of real news. The zero hypotheses, which claimed, that there is no statistically significant difference between the values of evaluation measures prec_real and rec_real in examined models, were rejected at the 0.001 significance level (prec_real: *G-G Epsilon* = 0.596, *H-F Epsilon* = 0.675, *adj.p* < 0.001; rec_real: *G-G Epsilon* = 0.842, *H-F Epsilon* = 0.975, *adj.p* < 0.001). The impact of the between-groups factor *deep* has not also been proven in this case ($p > 0.05$). It means that the performance of the models (prec_real/rec_real) does not depend on the interaction of the factors (*model* × *deep*).

The model merge(1,4) reached the highest quality from the evaluation measures, prec_real a rec_real point of view (Table 5). Only one homogeneous group was identified from the multiple comparisons for prec_real (Table 5A). PosF(1,4), TfIdf(1,4) and merge (1,4) reached the same quality of the model's predictions ($p > 0.05$). The statistically significant differences in the quality of the model's predictions (Table 5A) were identified between the PosfIdf(1,4) model and other models ($p < 0.05$).

Two homogenous groups (PosfIdf(1,4), PosF(1,4)) a (PosF(1,4), TfIdf(1,4)) were identified from the evaluation measure rec_real point of view (Table 5B). The statistically significant differences (Table 5B) were identified between the model merge(1,4) and other models ($p < 0.05$).

## Time efficiency

The time efficiency of the proposed techniques was evaluated in the second phase of the analysis. Time efficiency (*time*) was analysed in dependence on within-group factor and between-groups factor and their interaction. The models represented the examined levels of a within-group factor, and the decision tree depths represented the between-groups factor. The modified tests for repeated measures were again applied to verify the time efficiency of the proposed models. The values of *Epsilon* were identical and significantly

**Table 6 Bonferroni adjustment for the time: model.**

| Model | Time Mean | 1 | 2 | 3 |
|---|---|---|---|---|
| PosfIdf(1,4) | 5.354 | | **** | |
| TfIdf(1,4) | 15.527 | **** | | |
| PosF(1,4) | 17.908 | **** | | |
| merge(1,4) | 226.108 | | | **** |

**Note:**
**** Homogeneous groups ($p > 0.05$).

smaller than one for both corrections (*Epsilon* < 0.34). The zero hypothesis, which claimed that there is no statistically significant difference in time between the examined models, was rejected at the 0.001 significance level (time: *G-G Epsilon* = 0.336, *H-F Epsilon* = 0.366, *adj.p* < 0.001). Similarly, the zero hypothesis, which claimed that the time efficiency (*time*) does not depend on the interaction between the within-group factor and between-groups factor, was also rejected at the 0.001 significance level (*model × deep*). Factor *deep* has a significant impact on the time efficiency of the examined models.

Only one homogenous group based on time was identified from the multiple comparisons (Table 6). PosF(1,4) and TfIdf(1,4) reached the same time efficiency ($p > 0.05$). Statistically significant differences in time (Table 6) were identified between the merge(1,4) model and other models ($p < 0.05$), as well as between the PosfIdf(1,4) model and other models ($p < 0.05$). As a result, PosfIdf(1,4) model can be considered the most time effective model, while the merge(1,4) model was considered the least time-effective one.

Four homogeneous groups were identified after including between-group factor *deep* (Table 7). Models PosfIdf(1,4) with depth 6-10 and TfIdf(1,4) with depth six have the same time effectivity ($p > 0.05$). The models TfIdf(1,4) and PosF(1,4) have the same time efficiency for all depths ($p > 0.05$). The models merge(1,4) with depth 6-8 ($p > 0.05$) and models merge(1,4) with the depth 7-10 ($p > 0.05$) were less time effective.

## DISCUSSION

The paper analysed a unique dataset of the freely available fake and true news datasets written in English to evaluate if the POS tags created from the n-grams could be used for a reliable fake news classification. Two techniques based on POS tags were proposed and compared with the performance of the reference TF-IDF technique on a given classification task from the natural language processing research field.

The results show statistically insignificant differences between the PosF and TF-IDF techniques. These differences were comparable in all observed performance metrics, including accuracy, precision, recall and f1-score. Therefore, it can be concluded that morphological analysis can be applied to fake news classification. Moreover, the charts of descriptive statistics show TF-IDF technique reaches better results, though statistically insignificant.

It is necessary to note for completeness that the statistically significant differences in observed performance metrics were identified between the morphological technique

| Table 7 | Bonferroni adjustment for the time: deep × mode. | | | | | |
|---|---|---|---|---|---|---|
| **Deep** | **Model** | **Time mean** | **1** | **2** | **3** | **4** |
| 6 | PosfIdf(1,4) | 5.013 | **** | | | |
| 7 | PosfIdf(1,4) | 5.176 | **** | | | |
| 8 | PosfIdf(1,4) | 5.368 | **** | | | |
| 9 | PosfIdf(1,4) | 5.577 | **** | | | |
| 10 | PosfIdf(1,4) | 5.637 | **** | | | |
| 6 | TfIdf(1,4) | 14.343 | **** | **** | | |
| 7 | TfIdf(1,4) | 15.334 | | **** | | |
| 8 | TfIdf(1,4) | 15.717 | | **** | | |
| 6 | PosF(1,4) | 15.913 | | **** | | |
| 9 | TfIdf(1,4) | 16.083 | | **** | | |
| 10 | TfIdf(1,4) | 16.160 | | **** | | |
| 7 | PosF(1,4) | 17.624 | | **** | | |
| 8 | PosF(1,4) | 18.230 | | **** | | |
| 9 | PosF(1,4) | 18.828 | | **** | | |
| 10 | PosF(1,4) | 18.946 | | **** | | |
| 6 | merge(1,4) | 214.167 | | | **** | |
| 8 | merge(1,4) | 221.315 | | | **** | |
| 7 | merge(1,4) | 222.547 | | | **** | |
| 9 | merge(1,4) | 232.643 | | | | **** |
| 10 | merge(1,4) | 239.867 | | | | **** |

**Note:**
**** Homogeneous groups ($p > 0.05$)

PosfIdf and TF-IDF. The reason is that the PosfIdf technique includes the ratio of the relative frequency of POS tags and inverse document function. This division by the number of documents in which the POS tag was observed caused weak results of this technique. It is not surprising, whereas the selected 33 POS tags were included in almost all the dataset documents. Therefore, the value of inverse document frequency was very high, which led to a very low value of the ratio. However, the failure of this technique does not diminish the importance of the findings that applied morphological techniques are comparable with the traditional reference technique TF-IDF. The aim to find a morphological technique, which will be better than TF-IDF, was fulfilled in the case of the PosF technique.

The Merged TF-IDF and PosF technique was included in the experiment to determine whether it is possible to improve the reference TF-IDF technique using POS tags. Considering the final performance measures, mainly precision, it can be concluded that they are higher. It means that the applied techniques of morphological analysis could improve the precision of the TF-IDF technique. However, it has not been proven that this improvement is statistically significant.

The fact that the reference TF-IDF technique had been favoured in the presented experiment should be considered. In other words, removing the stop words from the input

**Table 8 Comparison of similar methods for fake news classification from text.**

| Methods for vector creation from text | Additional features | Classifier | Max Precision | Authors |
|---|---|---|---|---|
| Bag of word, Word2Vec, GloVe | domain names, authors, etc. | Long Short-Term Dependency (NN) | 89.19 | *Deepak & Chitturi (2020)* |
| GloVe | visual image features, hidden pattern extraction capabilities from text | Hierarchical Attention Network | 95.68 | *Meel & Vishwakarma (2021)* |
| Tf-Idf | morphological analysis, n-grams | Decision Trees | 83.48 | This research |

vector of the TF-IDF technique increased the classification accuracy. On the other hand, removing stop words is not suitable for the techniques based on the POS tags because their removal can cause losing important information about the n-gram structure. This statement is substantiated by comparing the values of accuracy for individual n-grams (Fig. 3). PosF technique achieved better results for 2-grams, 3-grams, 4-grams than for unigrams. Contrary, the stop words did not have to be removed from the input vector of the TF-IDF technique. However, the experiment aimed to compare the performance of the proposed improvements with the best prepared TF-IDF technique.

The time efficiency of the examined techniques was evaluated simultaneously with their performance. The negligible differences between the time efficiency of the TF-IDF and PosF techniques can be considered most surprising. Although the PosF technique uses only 33 POS tags compared to the large vectors of tokens/words in TF-IDF, the time efficiency is similar. The reason is that the POS tags identification in the text is more time-consuming than tokens identification. On the other hand, the merged technique with the best performance results was the most time-consuming. This finding was expected because the merged vector calculation requires calculating and joining the TF-IDF and PosF vectors.

The compared classification models for fake and true news classification are based on the relative frequencies of the occurrences of the morphological tags. It is not important which morphological tags were identified in the rules (nodes of the decision tree) using given selection measures. At the same time, the exact border values for the occurrence of morphological tags can also be considered unimportant because the more important fact is that such differences exist, and it is possible to find values of occurrences of morphological tags, which allow classifying fake and true news correctly.

The realized set of experiments is unique in the meaning of the proposed preprocessing techniques used to prepare the input vectors for classifiers. The decision was to use as simple classifiers as possible, thus decision trees, because of their ability to easily interpret the obtained knowledge. In other words, decision trees provide additional information, which POS tags and consequent n-grams are important and characteristic for the fake news and which for real news. On the other hand, it should be emphasized that their classification precision is worse than other types of classifiers like neutral networks.

Table 8 shows the comparison of similar methods for fake news classification (*Deepak & Chitturi, 2020*; *Meel & Vishwakarma, 2021*). The classification models reached a higher

accuracy. However, the authors reached these results using additional secondary features. The application of neural networks is the second important difference, which led to the higher classification performance.

However, as mentioned earlier, the experiment described in this paper was focused on assessing the suitability of the n-grams and POS-tags for pre-processing of input vectors. The priority was not to reach the best performance measures.

The selection of simple machine learning technique, like decision trees, can be considered the limitation of the research presented in this article. However, the reason why this technique was selected related to the parallel research, which was focused on the finding of the most frequent n-grams in fake and real news and their consequent linguistic analysis. This research extends the previous one and tries to determine if the POS tags and n-grams can be further used for fake news classification.

It is possible to assume that the morphological tags can be used as the input to the fake news classifiers. Moreover, the pre-processed datasets are suitable for other classification techniques, improving the accuracy of the fake news classification. It means that whether the relative frequencies of occurrences of the morphological tags are further used as the input layer of the neuron network or added to the training dataset of other classifiers, the found information can improve the accuracy of that fake news classifier.

## CONCLUSIONS

Despite several authors' statements that the morphological characteristics of the text do not allow fake news classification with sufficient accuracy, the realised experiment proved that the selected morphological technique is comparable with the traditional reference technique TF-IDF widely used in the natural language processing domain. The suitability of the techniques based on the morphological analysis has been proven on the contemporary dataset, including 1,100 labelled real and fake news about the Covid-19.

The experiment confirmed the validity of the newly proposed techniques based on the POS tags and n-grams against the traditional technique TF-IDF.

The article describes the experiment with a set of pre-processing techniques used to prepare input vectors for data mining classification task. The overall contribution of the proposed improvements was expressed by the characteristic performance measures of the classification task (accuracy, precision, recall and f1-score). Besides the variables defined by the input vectors, the hyperparameters max_depth and n-gram length were examined. K-fold validation was applied to consider the random errors. The global null hypotheses were evaluated using adjusted tests for repeated measures. Subsequently, multiple comparisons with Bonferroni adjustment were used to compare the models. Various performance measures ensured the robustness of the obtained results.

The decision trees were chosen to classify fake news because they create easily understandable and interpretable results compared to other classifiers. Moreover, they allow the generalisation of the inputs. An insufficient generalisation can cause overfitting, which leads to the wrong classification of individual observations of the testing dataset.

Different values of the parameter of maximal depth were researched to obtain the maximal value of precision. This most suitable value of the parameter was different for each of the proposed techniques. Therefore, the statistical evaluation was realised considering the maximal depth.

Besides the fact, the statistically significant difference has not been found, the proposed techniques based on the morphological analysis in combination with the created n-grams are comparable with traditional ones, for example, with the TF-IDF used in this experiment. Moreover, the advantages of the PoSF technique can be listed as follows:

- A smaller size of the input vectors. The average number of vectors elements was 69 775, while in the case of TF-IDF, it was 817 213.3 (Table 2).
- A faster creating of the input vector.
- A shorter training phase of the model.
- More straightforward and more understandable model. The model based on the PosF technique achieved the best results in smaller maximal decision tree depth values.

The possibility of using the proposed techniques based on POS tags on the classification of new yet untrained fake news datasets is considered the last advantage of the proposed techniques. The reason is that the TF-IDF works with the words and counts their frequencies in fake news. However, the traditional classifiers can fail to correctly classify fake news about a new topic because they have not yet trained the frequencies of new words. On the other hand, the PosF technique is more general and focuses on the primary relationships between POS tags, which are probably also similar in the case of new topics of fake news. This assumption will be evaluated in future research.

The current most effective fake news classification is based predominantly on neural networks and a weighted combination of techniques, which deal with news content, social context, credit of a creator/spreader and analysis of target victims. It clear that decision tree classifiers are not so frequently used. However, this article focused only on a very narrow part of the researched issues, and these classifiers have been used mainly for easier understanding of the problem. Therefore, future work will evaluate the proposed techniques in conjunction with other contemporary fake news classifiers.

### Funding
This work was supported by the Scientific Grant Agency of the Ministry of Education of the Slovak Republic (ME SR) and Slovak Academy of Sciences (SAS) under the contract No. VEGA-1/0792/21, also by the scientific research project of the Czech Sciences Foundation Grant No:19-15498S and by the Slovak Research and Development Agency under the contract no. APVV-18-0473. There was no additional external funding received for this study. The funders had no role in study design, data collection and analysis, decision to publish, or preparation of the manuscript.

## Grant Disclosures

The following grant information was disclosed by the authors:

Scientific Grant Agency of the Ministry of Education of the Slovak Republic (ME SR) and Slovak Academy of Sciences (SAS): VEGA-1/0792/21.

Czech Sciences Foundation: 19-15498S.

Slovak Research and Development Agency: APVV-18-0473.

## Competing Interests

The authors declare that they have no competing interests.

## Author Contributions

- Jozef Kapusta conceived and designed the experiments, performed the experiments, analyzed the data, performed the computation work, prepared figures and/or tables, and approved the final draft.
- Martin Drlik conceived and designed the experiments, performed the experiments, authored or reviewed drafts of the paper, and approved the final draft.
- Michal Munk conceived and designed the experiments, performed the experiments, analyzed the data, prepared figures and/or tables, and approved the final draft.

## Data Availability

The raw measurements and the source code for created models and measurements are available in the Supplemental Files.

## Supplemental Information

Supplemental information for this article can be found online at http://dx.doi.org/10.7717/peerj-cs.624#supplemental-information.

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
