# Peer review of "Using of n-grams from morphological tags for fake news classification"

_PeerJ Computer Science, doi:10.7717/peerj-cs.624_

## Round 0.1 · original submission · Major Revisions

In the light of the given comments, the authors are advised to revise the manuscript and provide three files:
1. Revised manuscript with highlighted changes
2. Revised manuscript, clean copy
3. Point to point rebuttal letter for reviewers comments.

Reviewer 1 ·

Basic reporting

The paper describes the Using of N-Grams from Morphological Tags for Fake News Classification. The paper is well written and presented well. Although, there are some of the comments, which need to be considered.
In abstract, author didn’t mention about the rationale behind the proposed system. Also, abstract has vague information, which need to be replaced with the important information. Also, it is necessary to add simulation parameters in the same section.
Introduction is too generic and is lacking important information about the N-Grams from Morphological Tags and proposed scheme. Why the author proposed such scheme is missing in the introduction. Also, it is necessary to add contribution and advantages of the proposed scheme at the end of the proposed scheme.
Background and related work is missing drawbacks of the existing scheme. Also, author of the paper has done some old survey. It is highly recommended to add latest references from reputable journals.
Proposed scheme section is missing an overview of the proposed scheme and also it is necessary to add overflow diagram in the same section. In addition, figures are not very clear. Please re-draw figures and add important information because you have mentioned about various systems but you didn’t include it in the explanation of the proposed figure diagram.
Results and evaluation is missing various terminologies. It is highly recommended to add the working of the implementation in algorithm. In addition, which parameters are considered and why they are considered is necessary to be added in this section.
Author should compare their scheme with existing schemes.
Also, results are not enough to defend the proposed scheme. Please add more results and discussion.
Conclusion need to revise as it looks like the summary of the paper. Also, please add latest references.

Experimental design

Results and evaluation is missing various terminologies. It is highly recommended to add the working of the implementation in algorithm. In addition, which parameters are considered and why they are considered is necessary to be added in this section.
Author should compare their scheme with existing schemes.
Also, results are not enough to defend the proposed scheme. Please add more results and discussion.
Conclusion need to revise as it looks like the summary of the paper. Also, please add latest references.

Validity of the findings

no comment

Additional comments

The paper describes the Using of N-Grams from Morphological Tags for Fake News Classification. The paper is well written and presented well. Although, there are some of comments, which need to be considered.
In abstract, author didn’t mention about the rationale behind the proposed system. Also, abstract has vague information, which need to be replaced with the important information. Also, it is necessary to add simulation parameters in the same section.
Introduction is too generic and is lacking important information about the N-Grams from Morphological Tags and proposed scheme. Why the author proposed such scheme is missing in the introduction. Also, it is necessary to add contribution and advantages of the proposed scheme at the end of the proposed scheme.
Background and related work is missing drawbacks of the existing scheme. Also, author of the paper has done some old survey. It is highly recommended to add latest references from reputable journals.
Proposed scheme section is missing an overview of the proposed scheme and also it is necessary to add overflow diagram in the same section. In addition, figures are not very clear. Please re-draw figures and add important information because you have mentioned about various systems but you didn’t include it in the explanation of the proposed figure diagram.
Results and evaluation is missing various terminologies. It is highly recommended to add the working of the implementation in algorithm. In addition, which parameters are considered and why they are considered is necessary to be added in this section.
Author should compare their scheme with existing schemes.
Also, results are not enough to defend the proposed scheme. Please add more results and discussion.
Conclusion need to revise as it looks like the summary of the paper. Also, please add latest references.

·

Basic reporting

The paper experimentally evaluates the potential of the common use of n-grams and POS tags for the correct classification of fake and true news. Continuous sequences of n items from a given sample of POS tags (n-grams) were analyzed. The techniques based on POS tags were proposed and used in order to meet this aim. These techniques were compared with the standardized reference TF-IDF technique to evaluate their main performance characteristics. Simultaneously, the question, whether the TF-IDF technique can be improved using POS tags, was researched in detail. All techniques have been applied in the pre-processing phase on different groups of n-grams. The resulted datasets have been analyzed using decision tree classifiers. The paper is well written and here I will suggest some points to improve the quality of the paper.

Experimental design

The flow of the experiment is not clear author should write concisely but make it clear and easy to understand.

Training and testing records shown be mention in a table for each target class.

Validation should be done by authors

Validity of the findings

Results are significant

Additional comments

The author has preset work with quality and the suggested point can improve the quality of the manuscript.

Typos and grammar needed to be check throughout the paper.
is there any results validation technique used by the author?
The quality of the figures should be improved.

---

## Round 0.2 · Minor Revisions

Based on the reports from the reviewers, It is highly advised to address the reviewer's concerns as required by the reviewer.

Reviewer 1 ·

Basic reporting

N/A

Experimental design

N/A

Validity of the findings

N/A

Additional comments

Upload the updated response to the reviewer report with section number, page number, and line number that should clearly show the reflected changes.

---

## Round 0.3 · Minor Revisions

Kindly add a Table for comparison of your work with the previous work to increase the readability of the paper. All other comments are resolved.
Thank you

---

## Round 0.4 · accepted · Accept

Based on the revisions made concerning reviewers and editor's comments, now the manuscript is accepted. Thank you for your time and effort for modifications.